# Investigating the Effect of Compassion Fade in Volunteer Tourism

**DOI:** 10.3390/bs12090319

**Published:** 2022-09-04

**Authors:** Sae-Mi Lee

**Affiliations:** The Center for Regional Development, Chonnam National University, Gwangju 61186, Korea; emailme6@naver.com or sm.lee@jnu.ac.kr

**Keywords:** compassion fade, volunteer tourism, nature-based tourism, emotional involvement, credibility, attitudes, behavioral intention

## Abstract

This study investigated the relationship between the type of volunteer tourism (human vs. flora vs. fauna) and the type of message (individual with no statistic vs. individual with small statistic vs. individual with large statistic) and potential tourists’ attitudes towards volunteer tourism and their intention to donate their time. To do so, this study conducted a between-subject 3 × 3 factorial design online experiment, where the influences of compassion fade on attitudes and behavioral intention to donate time for volunteer tourism, along with the impacts of positive affect, emotional involvement, and credibility, were examined. The results of the current study revealed that type of volunteer tourism and type of message do not affect attitude towards volunteer tourism and attitude towards the ad. Further analysis indicated that, among all three mediating variables, only positive affect mediated the relationship between type of volunteer tourism and attitude towards volunteer tourism, and all other hypotheses were not statistically significant. Moreover, the results indicated that there is a positive relationship between perceived ad credibility and attitude towards the ad, and also between perceived ad credibility and attitude towards volunteer tourism. The implications of these results are discussed based on the empirical findings.

## 1. Introduction

Currently, there are rising levels of forced displacement of people globally and not enough funds or assistance to sufficiently aid in the humanitarian crisis, according to the global humanitarian assistance report in 2021. While scholars have researched the ways in which message framing can impact monetary donations [1,2,3,4,5,6,7], less research has investigated the ways to increase an alternate form of assistance: donating time.

Volunteer tourism, which involves “aiding or alleviating the material poverty of some groups in society, the restoration of certain environments or research into aspects of society or environment” (p. 1) [8], is a meaningful and rewarding form of tourism since a tourist can gain self-development and altruistic experiences through participating in environmental conservation work or interactions with local communities [9,10]. As more and more organizations are advertising for this type of tourism, volunteer tourism seems to be gaining in popularity (e.g., the result of a google search “volunteer tourism” on 12 June 2022 returned 2,140,000,000 hits). However, previous empirical studies in the volunteer tourism field focused mainly on personal characteristics of volunteer tourists or perspective of community members at a particular destination on volunteer tourism [9,10,11]. In other words, scholars in the volunteer tourism context have neglected how and why individuals participate in volunteer tourism via marketing communication channels, such as social media posts and messages. Therefore, this research attempts to identify motivational factors of participating in volunteer tourism (or time donation) from a general traveler’s perspectives on an advertisement. This study’s approach helps practitioners and scholars in this field to market a volunteer tourism product and make it more attractive to general travelers, by focusing on the identified motivational factors for the advertising strategy.

Although volunteer tourism has been conceptually divided into community-based and nature-based [11], prior empirical research has been conducted within more of an environmental conservation than a community interaction context [11,12]. Some scholars suggest that there is a critical need for research on providing a way of investigating each type of volunteer tourism [13]. This is due in part to people’s differing motivations for participation, such as that community-based volunteer tourism participants may elicit different benefits and expected experiences than those of the nature-based programs.

Since there has been comprehensive literature on monetary donation for people and natural environment [3,4,5,6,14], this study examines the influence of type of message on time donation based on the compassion fade framework. Specifically, when confronted with one child who suffers from starvation, individuals often move their hearts and help the child with their donation [3]. However, if the number of children increases to two or three, both emotional and behavioral compassion will begin to wane, which is referred to as compassion fade (i.e., reduce supportive or helping behaviors) [3,4]. It is necessary to investigate whether or not compassion fade can be applicable to time spending of voluntary activities rather than money spending, such as volunteer tourism activities. Since time spending requires higher levels of efforts and involvements than monetary donations, the compassion fade framework should be applied to the volunteer tourism context with consideration of its core components, including emotional involvement, credibility, and positive affect, along with attitudes towards the message. To be specific, individuals’ assessment of money is less likely to be ambiguous and their perception of monetary constraints tends to be less elastic than those of time donation, such as volunteer tourism [12,14]. In addition, volunteer tourism provides participants with direct experiences that require high levels of involvement, emotions, and feelings with community members at a particular destination in need [13]. However, monetary donations require individuals to send money or click a donation button at home or work. These characteristics of volunteer tourism (or time donation) lead scholars to consider participants’ emotions and extra efforts for others in need. Furthermore, since monetary donations are relatively simpler than time donations (e.g., indirect support via a click button at home/work vs. direct support via interactions with people in need at a particular destination), individuals’ intention to participate in monetary donations is likely to be easily influenced by contextual or personal factors (i.e., income level or economic conditions) [8,9]. More interestingly, from a behavioral perspective, the intention to donate time tends to be perceived as a more responsible act than a monetary donation among individuals because time donations enable individuals to create positive attributions via the particular behavior [6,7]. For example, the perceived goal of monetary donation is associated with individuals’ economic utility, whereas the perceived goal of time donation is related to their emotional health and responsibility [13].

Prior research in the charitable donation field has focused primarily on monetary donation based on the perspective on compassion fade, neglecting time donation. Time donations can lead participants to build higher levels of social capital and loyalty towards a particular place than monetary donations. The characteristics of time donations make this study important (e.g., social sustainability). Since there are behavioral and psychological differences between monetary donation and time donation, it is necessary to apply the impact of compassion fade to the time donation context, such as volunteer tourism. The approach of this study enables scholars and practitioners to develop a psychological framework of time donation behaviors within the context of volunteer tourism. Therefore, this study investigates the relationship between the type of volunteer tourism (human vs. flora vs. fauna) and the type of message (individual with no statistic vs. individual with small statistic vs. individual with large statistic) and potential tourists’ attitudes towards volunteer tourism and their intention to donate their time. To answer this research question, we conducted a between-subject 3×3 factorial design online experiment, where the influences of compassion fade on attitudes and behavioral intention to donate time for volunteer tourism, along with the impacts of positive affect, emotional involvement, and credibility, were examined.

## 2. Literature Review

### 2.1. The Impact of Compassion Fade on Attitude and Behavioral Intention

According to the compassion fade framework, a group of victims elicit weaker emotional reactions or feelings than a single victim [6]. Kogut and Ritov’s study [2] indicated that the processing of information with regards to a group of victims might fundamentally differ from the processing of information associated with a single victim. Their research addressed that an individual is more likely to feel less compassion and distress when he/she considers a group of victims than a single victim, which leads to a weaker intention to help the victims. Research shows that a group of victims arouses less positive affect in participants, which leads to less charitable action because group victims are processed differently from individual victims [6,15].

However, the valuation of human lives changes when put in the context of a group, whereas the individual shows less willingness to act pro-socially for the victims as the total population of victims increases, even when the number of lives being saved does not change [3,4]. In other words, people are more likely to help the plight of “the one” identified victim, but become psychically numb when that one is part of a larger problem [4].

Compassion fade can also emerge when a victim is a non-human animal. Smith, Faro, and Burson [16] considered a natural environment as a target of aid and their study indicated that the processing of compassion fade in the human context could be applied to the animal context. In addition, we predict that the degree of compassion fade regarding non-human beings may differ across contexts, such as fauna and flora.

Further research has accounted for compassion fade with numerous, motivational, cognitive, and affective frameworks [3]. For instance, a decision maker tends to show stronger compassion to one victim particularly since a group may elicit an inherently weaker emotional response than does a single individual [4]. This is because groups of people tend to be perceived as less cordial [16]. Even though individuals have well practiced how to take another person’s standpoint, which in general encourages prosocial behavior, they feel it is more difficult to take a group’s perspective [17]. From a motivational perspective, Cameron and Payne [14] have suggested that people have preemptive down-regulations of their affective responses to a helping situation in which they know that they will be asked to aid multiple people. Based on this notion, this study establishes the following hypotheses regarding compassion fade.

**H1a,** **H1b,** **H1c:**
*In line with the compassion fade framework, as the number of victims increases, we predict for users to feel less compassion, which will lead to (a) a decrease in attitude towards volunteer tourism, (b) less emotional involvement, and (c) less positive affect.*


### 2.2. The Impact of Emotional States on Attitude and Behavioral Intention

Prior research has been conducted on how types of messages can influence attitudes, behavioral intention, and even actual behaviors [18]. During the message delivering process, emotional states can play an important role. For instance, research has indicated that types of messages result in different emotional responses to the message and its information and different attitudes towards the message and its information [18,19]. Furthermore, an emotion has served as a mediator of message effects, which play an indirect role to underlie psychological procedures affecting personal responses and helping to explain how and why message effects take place [20]. Prior researchers, for example, indicated that types of messages evoked distinct emotional responses, which, in turn, led to different levels of attitudes and types of behaviors towards the message [18].

An affective feeling (e.g., compassion, sympathy, empathy, or sadness) has been considered to motivate an individual to help others [21]. For example, Slovic [4] establishes a model considering mental attention and images as determinants of an affective feeling towards other people in agony. Furthermore, Dickert and Slovic [22] indicate that an individual’s sympathy towards a child who needs help was lesser when the individual made judgments from his/her memory than when the individual’s attention directly focuses on the child. Compassion towards the particular child was also reduced by distractor children in the visual aid [7]. Those affective feelings can motivate behaviors, such as decision making [23]. For example, in several decision-making situations, relying on emotion and affect as sources of behaviors is more likely to be an efficient way of navigating in an uncertain circumstance (e.g., volunteer tourism) (Kahneman, 2011). In line with this thinking, the following hypothesis is proposed:

**H2a,** **H2b,** **H2c:**
*In line with the compassion fade framework, as the number of victims increases, we predict for users to feel (a) less compassion, (b) less emotional involvement, and (c) less positive affect, which will lead to a decrease in attitude towards the ad.*


### 2.3. The Impact of Human Condition

Since humans are more likely to react stronger to similar beings, this study predicts that participants will have a stronger affect response to the human conditions as opposed to the non-human (flora and fauna) conditions.

**H3a,** **H3b,** **H3c:**
*For the human condition, we predict for users to feel (a) higher compassion, (b) higher emotional involvement, and (c) positive affect, which would lead to an increase in attitude towards volunteer tourism.*


**H4a,** **H4b,** **H4c:**
*For the human condition, we predict for users to feel (a) higher compassion, (b) higher emotional involvement, and (c) positive affect, which would lead to an increase in attitude towards the ad.*


### 2.4. The Impact of Credibility on Attitude and Behavioral Intention

Credibility refers to the positive characteristics of a message sender affecting acceptance of the message of a receiver, and it determines the message’s persuasiveness [24]. Furthermore, message evaluation, attitudes towards advertising, behavioral intention, and even actual behaviors are influenced by a receiver’s perceptions of credibility of advertising [25]. An advertisement with more-credible sources has been found to enhance more positive attitudes and lead to stronger behavioral intention than an advertisement with less-credible sources [24,25]. Prior research has identified trustworthiness as a potentially critical and enduring dimension of perception of credibility [25,26]. Trustworthiness is defined as a belief of an audience that a message sender offers information in an honest manner regardless of a motivation for deception or manipulation [24]. The process of developing trustworthiness of advertising is a critical component for advertising effects [27]. Based on this notion, this study establishes the following hypotheses regarding credibility.

**H5,** **H6:**
*There would be a positive relationship between perceived ad credibility and attitude towards the ad (H5) and attitude towards volunteer tourism (H6).*


## 3. Method

We conducted an online experiment using a 3 (type of volunteer tourism: flora vs. fauna vs. human) × 3 (message type: no statistic vs. low statistic vs. high statistic) between-subject design. Participants (*n* = 162) were randomly assigned to one of nine possible conditions in which they were asked to view an advertisement for one of three possible types of volunteer tourism. After viewing the advertisement, participants completed a survey that measured their levels of compassion, attitude towards the ad, attitude towards volunteer tourism, emotional involvement, positive affect, credibility of the ad, behavioral intention to donate time, and actual intention to donate time.

### 3.1. Participants

One hundred and sixty-two people from North America were recruited to participate in the study using the crowdsourcing web site Amazon’s Mechanical Turk [28], which was approved by the Institutional Review Board at the university where the study was approved. All participants provided implied consent on an online questionnaire before participation. The increasing number of volunteer tourism products in the United States has made Americans’ intention and behavior to donate time outcomes of this research. In particular, issues concerning natural and/or man-made disasters and polarization are of particular interest to Americans who care about well-being of the global world [29]. They consider volunteer tourism as improved options for combating global inequalities as well as pursuing global development, making positive changes abroad [10]. However, scholars and practitioners need to understand the motivational determinants of a first participant’s time donation since the volunteer tourism industry has relied heavily on travelers’ revisit and loyalty [10,12].

Amazon’s Mechanical Turk enables scholars to collect culturally and geographically diverse samples in the United States as the unit of analysis in this study was general travelers in the United States. In addition, via random sampling, Amazon’s Mechanical Turk helps scholars to reduce sample selection bias (e.g., skewness in gender, age, education levels, or income) [28]. To maximize the participants’ involvement and engagement in this experiment, the author of this study offered a monetary incentive (i.e., USD 0.50) to those who completed the experiment. The sample distribution was 54% male (*n* = 87) with a mean age of 36.29 (SD = 10.73). When asked to self-report their race, 78% reported White/Caucasian (*n* = 126), 8% reported Black/African American (*n* = 14), 8% reported Asian/Asian American (*n* = 14), 5% reported Hispanic/Latino/Latina (*n* = 7), and 1% reported other (*n* = 1).

### 3.2. Stimuli

In this study, participants viewed one of nine possible advertisements for volunteer tourism. Two countries were used (Sri Lanka or Cambodia) so that study results would be generalizable to a variety of possible advertisements, otherwise, aside from the experimental manipulations (see below), attributes of the ads were held consistent throughout the conditions.

### 3.3. Data Analysis

This study used IBM SPSS Statistics 28.0 to perform descriptive analysis, reliability analysis (i.e., Cronbach alpha coefficient), one-way ANOVA (Analysis of Variance), two-way ANOVA, logistic regression, and single linear analysis to test research hypotheses in a rigorous manner.

## 4. Experimental Manipulation

### 4.1. Type of Volunteer Tourism

As outlined above, participants were exposed to one of three possible volunteer tourism advertisements featuring either flora (i.e., plants, vegetation), fauna (i.e., animals), or humans (i.e., children). For the flora ad condition, a photo of a freshly planted tree sapling accompanied by text that read, “Meet a duter tree. When you volunteer abroad, you could help this tree reach its potential. This tree grows in the lush central highlands of Sri Lanka [Cambodia]. It produces units of oxygen and provides shade in an otherwise sunny area”, with the numeracy manipulated in each condition (see below). The fauna condition featured a photo of an elephant. The tag line included in this manipulation said, “Meet Koki. When you volunteer abroad, you could help her reach her potential. Koki lives in the lush central highlands of Sri Lanka [Cambodia]. She enjoys sunbathing and eating fresh shoots of vegetation.” In the human condition, the text was accompanied by a photo of a young, smiling boy with dark features. The text said, “Meet Sanga. When you volunteer abroad, you could help him reach his potential. Sanga lives in the lush central highlands of Sri Lanka [Cambodia]. He enjoys science and likes playing soccer with his friends.” All ads were accompanied by a supporting text including a mock website link and quote from Helen Keller that said, “Alone we can do so little, together we can do so much.” Additionally, all ads featured a volunteering for change company logo.

### 4.2. Message Type

Each of the advertisement’s tag lines included a manipulation of the numeracy. In the no-statistic condition, the figure was identified by name or type (i.e., Sanga, Koki, duter tree). In the low-statistic condition, the figure was still identified by name or type, but it was also accompanied by the phrase “you could help him and other children just like him” (or for subsequent conditions: her/elephant; it/tree). In the high-statistic condition, again the figure was identified by name or type and had a larger statistic that read “you could help him and millions of other children just like him” (or for subsequent conditions: her/elephant; it/tree).

### 4.3. Mediating Variables

*Compassion fade*. To measure compassion fade, one Likert-type item (1 = “not at all,” 7 = “very much”) was adapted from prior research [29]. Participants were asked to indicate their agreement with the statement “I felt sympathy and compassion towards the child [animal/plant] in the ad.”

*Emotional involvement*. As for emotional involvement, four Likert-type items (1 = “not at all,” 7 = “very much”) were adapted from prior research [30]. The participants were asked to indicate their level of agreement with four statements about the advertisement viewed during the study such as “I found myself responding strongly to the advertisement” and “I got involved with the information and content on the advertisement.” The items had acceptable inter-item reliability, so an index was formed (Cronbach’s α = 0.938; *M* = 19.41, *SD* = 7.554).

*Credibility*. Six Likert-type items (1 = “not at all,” 7 = “very much”) were adapted from prior research [31] to measure the participants’ impressions of credibility of the advertisements. They were asked to indicate their level of agreement with statements about the advertisement such as “I trust the information in the advertisement”, “I found the stories featured on the advertisement to be reliable”, and “I found the stories featured on the advertisement to be believable.” The six items had acceptable inter-item reliability, so an index was formed (Cronbach’s α = 0.959; *M* = 30.44, *SD* = 8.154).

### 4.4. Dependent Variables

*Attitude towards the ad*. To gauge participants’ attitudes towards the ad, participants were asked to complete eleven Likert-type items (1 = “strongly disagree,” 7 = “strongly agree”), adapted from Ivory and Kalyanaraman [32]. Participants rated the advertisements for items such as “appealing”, “interesting”, and “high-quality.” The eleven items had adequate inter-item reliability, so an index was formed (Cronbach’s α = 0.958; *M* = 45.61, *SD* = 14.49).

*Attitude towards volunteer tourism*. Three Likert-type items (1 = “strongly disagree”, 7 = “strongly agree”) were used to measure participants’ attitude towards volunteer tourism, adapted from prior research [1]. For example, the participants were asked to indicate their level of agreement with statements such as “I believe that volunteer tourism helps to improve the lives of people”, “I would regularly participate in volunteer tourism if I had enough time available”, and “I believe that, in general, volunteer tourism is a meaningful way to help people.” For this scale, the wording was altered to match the type of volunteer tourism stimuli they viewed during the study (human vs. flora vs. fauna). The three items had acceptable inter-item reliability, so an index was formed (Cronbach’s α = 0.676; *M* = 15.629, *SD* = 3.383).

*Positive affect*. Four Likert-type items (1 = “not at all”, 7 = “very much”) measuring positive affect (e.g., cheerful, happy) were adapted from prior research [33]. The four items had adequate inter-item reliability, so an index was formed (Cronbach’s α = 0.953; *M* = 17.69, *SD* = 6.448).

*Behavioral intention to donate time*. To measure the participants’ intention to donate their time, participants were asked to imagine that they had the opportunity to participate in a volunteer tourism program. Details were given about the fictitious company’s credentials and how their participation in the program would be used to alleviate the situation in the country of their choosing. Participants then responded “yes” or “no” to the question “would you want to participate in a program like this?” Then, they were asked to indicate how many hours, up to 100 h, that they were willing to donate to the program.

*Actual intention to donate time*. To obtain a more accurate measure of participants’ intention to donate their time, we used a behavior measure that prompted participants to take an action that would seemingly have implications beyond the scope of the study. The participants read a prompt about a fictitious foundation that was hosting a scholarship raffle to fund people to travel abroad to volunteer. After being assured that their contact information would not be shared with anyone except the organization and would only be used in the event that they won, they were asked if they would like to add their contact information to the scholarship raffle with a “yes” or “no” response.

*Other Measures*. For the purpose of compiling descriptive statistics, participants were asked to report their age, gender, and race before taking part in the survey.

## 5. Results

*Compassion fade.* A one-way ANOVA with type of message as the independent variable and compassion fade as the dependent variable was not significant for manipulation checks: source type, *F* (2,161) = 2.323, *p* > 0.05. There were not significant differences between participants’ level of compassion among the three message conditions.

*Emotional involvement.* A one-way ANOVA with type of message as the independent variable and emotional involvement as the dependent variable was not significant for manipulation checks: source type, *F* (2,161) = 0.664, *p* > 0.05. There were not significant differences between participants’ emotional involvement among the three message conditions.

The study’s main research question asked about the relationship between type of volunteer tourism (human vs. flora vs. fauna) and type of message (individual with no statistic vs. individual with small statistic vs. individual with large statistic) and potential tourists’ attitudes towards volunteer tourism and attitude towards the ad and intention to donate time.

The results of ANOVA with type of volunteer tourism and type of message as fixed factors and attitude towards volunteer tourism as the dependent variable revealed that the overall model was significant, *F* (8,152) = 2.667, *p* < 0.05, R^2^ = 0.122. The analysis showed a significant main effect for the two-way interaction between dimensions, *F* (4,152) = 3.230, *p* < 0.05, R^2^ = 0.078, but no significant effect was found for the type of volunteer tourism, *F* (2,152) = 1.945, and type of message, *F* (2,152) = 2.262. The results of ANOVA with type of volunteer tourism and type of message as fixed factors and attitude towards the ad as the dependent variable revealed that the overall model was not significant, *F* (8,152) = 0.383. Therefore, no significant main affect and no significant two-way interaction between dimensions were found. Therefore, type of message/type of volunteer tourism did not affect attitude towards volunteer tourism and attitude towards the ad. Therefore, the results indicated that type of volunteer tourism and type of message do not affect attitude towards volunteer tourism and attitude towards the ad.

The results of an analysis of logistic regression with type of volunteer tourism and type of message as covariates and actual charitable behavior as the dependent variable indicated that type of volunteer tourism (EXP (B) = 0.778, Wald (1) = 1.115, *p* = 0.291) and type of message (EXP (B) = 0.863, Wald (1) = 0.374, *p* = 0.541) were not significant predictors of participants’ actual charitable behavior. Furthermore, the results of an analysis of logistic regression with type of volunteer tourism and type of message as covariates and charitable behavioral intention as the dependent variable indicated that type of volunteer tourism (EXP (B) = 0.959, Wald (1) = 0.042, *p* = 0.838) and type of message (EXP (B) = 0.882, Wald (1) = 0.375, *p* = 0.541) were not significant predictors of participants’ charitable behavioral intention. Therefore, the results indicated that there is no relationship between type of volunteer tourism/type of message and actual charitable behavior and charitable behavioral intention.

H1a predicted that as the number of victims increased, participants would feel less compassion, which would lead to a decrease in attitude towards volunteer tourism, and H3a predicted that for the human condition, participants would show higher compassion, which would lead to an increase in attitude towards volunteer tourism. In order to examine these hypotheses, an ANOVA was conducted with compassion fade as a covariate, along with type of volunteer tourism and type of message as independent variables and attitude towards volunteer tourism as the dependent variable. The overall model was significant, *F* (9,152) = 17.053, *p* < 0.05, R^2^ = 0.502. The analysis showed a significant main effect for a two-way interaction between dimensions, *F* (4,152) = 2.633, *p* < 0.05, R^2^ = 0.065. Compassion fade was also shown to be a significant predictor of attitude towards volunteer tourism, *F* (1,152) = 139.676, *p* < 0.05, R^2^ = 0.479. However, no significant main effects were found for (H1a) type of message, *F* (2,152) = 0.319, or (H3a) type of volunteer tourism, *F* (2,153) = 1.318. Therefore, H1a and H3a were not supported.

H1b predicted that as the number of victims increased, participants would show less emotional involvement, which would lead to a decrease in attitude towards volunteer tourism, and H3b predicted that for the human condition, participants would show higher emotional involvement, which would lead to an increase in attitude towards volunteer tourism. Again, an ANOVA was conducted with emotional involvement as a covariate, along with type of volunteer tourism and type of message as independent variables, and attitude towards volunteer tourism as the dependent variable. The overall model was significant, *F* (9,152) = 19.289, *p* < 0.05, R^2^ = 0.533. Emotional involvement was shown to be a significant predictor of attitude towards volunteer tourism, *F* (1,152) = 158.972, *p* < 0.05, R^2^ = 0.511. However, no significant main effects were found for (H1b) type of message, *F* (2,152) = 2.125, or (H3b) type of volunteer tourism, *F* (2,152) = 0.850 and there were no significant interactions, *F* (4,152) = 2.170. Therefore, H1b and H3b were not supported.

H1c predicted that as the number of victims increased, participants would feel less positive affect, which would lead to a decrease in attitude towards volunteer tourism, and H3c predicted that for the human condition, participants would feel more positive affect, which would lead to an increase in attitude towards volunteer tourism. Again, an ANOVA was conducted, with positive affect as a covariate, along with type of volunteer tourism and type of message as independent variables, and attitude towards volunteer tourism as the dependent variable. The overall model was significant, *F* (9,152) = 9.446, *p* < 0.05, R^2^ = 0.359. The analysis showed a significant main effect for (H3c) type of volunteer tourism, *F* (2, 152) = 10.303, *p* < 0.05, R^2^ = 0.119, such that participants in the human condition had more positive attitude towards volunteer tourism (*M* = 5.507, *SD* = 0.154) than those in the flora volunteer tourism condition (*M* = 4.574, *SD* = 0.151) and fauna conditions (*M* = 4.804, *SD* = 1.50). Positive affect was also shown to be a significant predictor of attitude towards volunteer tourism, *F* (1,152) = 49.808, *p* < 0.05, R^2^ = 0.247. Furthermore, a significant main effect for two-way interaction between dimensions, *F* (4,152) = 2.761, *p* < 0.05, R^2^ = 0.068, was found. No significant main effects were found for (H1c) type of message, *F* (2,152) = 1.779. Therefore, H3c was supported and H1c was not supported.

H2a predicted that as the number of victims increased, participants would feel less compassion, which would lead to a decrease in attitude towards the ad, and H4a predicted that for the human condition, participants would show higher compassion, which would lead to an increase in attitude towards the ad. In order to examine these hypotheses, an ANOVA was conducted with compassion fade as a covariate, along with type of volunteer tourism and type of message as independent variables and attitude towards the ad as the dependent variable. The overall model was significant, *F* (9,152) = 7.814, *p* < 0.05, R^2^ = 0.316. The analysis showed a significant main effect for (H4a) type of volunteer tourism, *F*(2, 152) = 4.001, *p* < 0.05, R^2^ = 0.050, such that participants in human volunteer tourism condition had less positive attitude towards the ad (*M* = 4.640, *SD* = 0.154) than those in the flora volunteer tourism condition (*M* = 5.291, *SD* = 0.157), and there was no difference between participants’ attitude towards the ad in human and fauna conditions (*M* = 4.973, *SD* = 1.53). Compassion fade was also shown to be a significant predictor of attitude towards the ad, *F* (1,152) = 58.059, *p* < 0.05, R^2^ =0.276. No significant main effects were found for (H2a) type of message, *F* (2,152) = 1.395, and there were no significant interactions, *F* (4,152) = 0.369. Therefore, H2a and H4a were not supported.

H2b predicted that as the number of victims increased, participants would show less emotional involvement, which would lead to a decrease in attitude towards the ad, and H4b predicted that for the human condition, participants would show higher emotional involvement, which would lead to an increase in attitude towards the ad. Again, an ANOVA was conducted, with emotional involvement as a covariate along with type of volunteer tourism and type of message as independent variables and attitude towards the ad as the dependent variable. The overall model was significant, *F* (9,152) = 9.377, *p* < 0.05, R^2^ = 0.357. Emotional involvement was also shown to be a significant predictor of attitude towards the ad, *F* (1,152) = 78.313, *p* < 0.05, R^2^ =.340. The analysis showed no significant main effect for (H4b) type of volunteer tourism, *F* (2,152) = 2.237, or (H2b) type of message, *F* (2,152) = 0.425. Furthermore, no significant main effect for the two-way interaction between dimensions was found. Therefore, H2b and H4b were not supported.

H2c predicted that as the number of victims increased, participants would feel less positive affect, which would lead to a decrease in attitude towards the ad, and H4c predicted that for the human condition, participants would feel more positive affect, which would lead to an increase in attitude towards the ad. Again, an ANOVA was conducted, with positive affect as a covariate, along with type of volunteer tourism and type of message as independent variables and attitude towards the ad as the dependent variable. The overall model was significant, *F* (9,152) = 9.419, *p* < 0.05, R^2^ = 0.358. Positive affect was shown to be a significant predictor of attitude towards ad, *F* (1,152) = 77.779, *p* < 0.05, R^2^ = 0.338. The analysis showed no significant main effect for (H4c) type of volunteer tourism, *F* (2,152) = 1.562, or (H2c) type of message *F* (2,152) = 0.945. Furthermore, no significant main effect for the two-way interaction between dimensions was found. Therefore, H2c and H4c were not supported.

H5 predicted that there would be a positive relationship between perceived ad credibility and attitude towards the ad. The result of single linear regression with perceived ad credibility as the independent variable and attitude towards the ad as the dependent variable indicated that there is a statistically significant positive relationship between perceived ad credibility (b = 0.640, *t*(160)= 11.144, *p* < 0.05) and attitude towards the ad. As participants perceived more ad credibility, they showed higher positive attitude towards the ad. Therefore, H5 was significant.

Furthermore, H6 predicted that there is a positive relationship between perceived ad credibility and attitude towards volunteer tourism. The result of linear regression with perceived ad credibility as the independent variable and attitude towards volunteer tourism as the dependent variable indicated that there is a statistically significant positive relationship between perceived ad credibility (b = 0.690, *t*(160) = 12.509, *p* < 0.05) and attitude towards volunteer tourism. As participants perceived more ad credibility, they showed higher positive attitude towards volunteer tourism. In sum, H6 was significant.

In sum, the results of the current study revealed that type of volunteer tourism and type of message do not affect attitude towards volunteer tourism and attitude towards the ad. Furthermore, there is no relationship between type of volunteer tourism and type of message and attitude towards the ad and attitude towards volunteer tourism. Further analysis indicated that, among all three mediating variables (e.g., compassion fade, emotional involvement, positive affect) only positive affect mediated the relationship between type of volunteer tourism and attitude towards volunteer tourism (H3c), and all other hypotheses were not statistically significant. Moreover, results indicated that there is a positive relationship between perceived ad credibility and attitude towards the ad and also between perceived ad credibility and attitude towards volunteer tourism. The implications of these results are discussed below.

## 6. Discussion

While the global humanitarian crisis continues to receive attention [34], research has asked how message framing can impact people’s monetary donations [1,2,3,5,6,7], yet less research has asked how appeals can be adapted to garner volunteers to donate their time instead of their money. Volunteer tourism or “voluntourism” is the act of people traveling to other countries to donate their time to help the community. Informed by compassion fade as an explanatory framework, the present study examined the effects of message type on participant attitudes and intention to donate time. Additionally, this project attempts to establish three distinct types of voluntourism: fauna, flora, and humans. The implications of the study’s findings are discussed below.

First, although voluntourism has been conceptually divided into two realms and studied in terms of the community-based and nature-based [11], this paper offers a conceptual frame to include flora (plants), fauna (animals), and humans as distinct forms of voluntourism, since people’s motivation, benefits, and expectations may vary greatly between the three distinct categories. As a step in the right direction, in the current study, when participants were randomly assigned to one of the three types of voluntourism, they indicated the category that corresponded to their stimuli most of the time (93%). In future studies, this relationship could be elucidated further by examining people’s expectations and motivation for each of the distinct categories, but this study offers an initial step to determining the different categorizations of this type of tourism.

Although compassion fade is usually studied with monetary donations, this study attempted to extend the literature to donations of time and attitudes towards such types of volunteering activities. Consistent with the compassion fade framework, research shows that unlike a group of people, a single individual is viewed differently psychologically. People respond to single individuals as a well-defined unit, which leads the viewer to process the information more and create a stronger impression about the individual as opposed to a group [7,35,36]. Furthermore, research shows that stronger affect feelings are evoked when people consider single victims as opposed to a group of victims; specifically, participants reported stronger feelings of distress and compassion, which lead to higher donations when examining a single identified victim than compared to a group of victims [2]. Therefore, in the current study, we examined if this effect could be studied when participants were asked to donate their time as opposed to their money to help victims. As similarly predicted in Västfjäll et al. [7] for charitable donations, we expected positive affect to be strongest for a single, identified individual. This hypothesis was not supported. Furthermore, in line with the compassion fade construct, we predicted that as the number of victims increased, the reported compassion would decrease, which would lead to a decrease in intention to donate time, just as it has been shown with monetary donations. This hypothesis was also not supported. We also could not find consistent results with participants’ attitude towards the ad or attitude towards volunteer tourism. However, we did find support for the credibility hypothesis, which bolsters the argument that the more credible the participant perceived the stimuli, the more favorable they reported their (a) feelings towards volunteer tourism and (b) feelings towards the ad. We suspect that we encountered such non-significant findings because the manipulation checks on the stimuli failed. Although these findings appear challenging, the possible limitations that could account for these results are discussed below, as well as future directions for this body of work.

While the results of the present study hold some theoretical and practical implications, several limitations should contextualize the findings. Overall, this study could not replicate many of the findings that have been found in the compassion fade literature. One reason this could have occurred is because our sample size was relatively small (*n* = 162). Several of the main effects were approaching significance and it would take further study power to determine if more of the study’s hypotheses are actually significant. Another possible reason we did not find many significant results could be that the manipulation was too subtle. In future studies, to make the numeracy manipulation more salient, we could emphasize the statistic either by presenting it separately from the advertisement or by building in a manipulation check to see if they could even recall the statistic that was shown to them. Moreover, this study also suffered measurement errors. Most notably, we measured participants’ intention to donate on a binary (yes/no) scale. In reality, if we had measured it on a Likert-type scale (1 = not at all; 7 = very likely), we could have run more dynamic statistical tests to examine the relationship between variables better, so this should be taken into account for future studies. Additionally, the stimuli were modeled after an actual ad, and showed neutrally framed wording to accompany a pleasant photo (i.e., a smiling boy; an elephant with perky ears). Future studies could consider manipulating the frame of the wording to see if a gain vs. loss framing could stimulate people with a call-to-action (i.e., if people do not volunteer their time, something adverse would happen to the people/plant/animal featured in the stimuli). Taking all of these concerns into consideration, this study has potential to be improved.

In a related vein, another plausible reason for finding inconsistencies across conditions could be that individual differences could account more for people’s willingness to travel and not necessarily measurement errors. According to a recent study by Hostelworld Global Traveler Report, nearly one-third of Americans (29%) have never been abroad, and they are about half as likely to travel to more than one country as compared to Europeans [37]. This would suggest that perhaps participants’ past international travel experience could impact the way they view and respond to the stimuli. Future studies should consider measuring for this construct to rule out this possible alternative explanation. Another way to gain perspective on this particular variable would be to replicate the current study, but instead of using poor, foreign nations, research could study the effects of psychological distance on participants’ willingness to donate their time. Therefore, would something geographically closer to the participant make a difference in their attitudes and intentions?

## 7. Conclusions

The purpose of prior research in compassion fade has been to primarily measure and predict individuals’ intention to donate money for others in need or for nature in danger, separately. However, compared to monetary donations, time donations, such as volunteer tourism, can bring more positive things to a particular destination and its communities, including economic (e.g., via travel) and social (e.g., via help) benefits. Thus, this empirical research attempted to contribute to the volunteer tourism and compassion fade literature with consideration of individuals’ intention and behavior to donate time for others in need and for nature in danger, simultaneously. To do so, this study was based on the psychological framework of compassion fade and applied its core variables (i.e., emotional involvement, credibility, and positive affect) to the volunteer tourism context. Although most of the findings of this study were insignificant, it lays a good foundation for future directions and studies. A larger sample size is needed to determine stronger relationships between variables and follow-up studies to examine the construct of psychological distance to participants’ willingness to travel are needed. As the humanitarian crises continue to grow, study findings suggest that there is not a quick and easy fix to the charitable donation problem facing the globe.

## Data Availability

The data presented in this study are available on request from the corresponding author.

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
