# Peer review of "Investigating the Effect of Compassion Fade in Volunteer Tourism"

_behavsci, 2022, doi:10.3390/bs12090319_

Round 1

Reviewer 1 Report

The topic is exciting, and the proposed objective of the study has significant practical implications. 

However, the consistency of the link between the supporting theoretical models and the social context is unclear to the readers. The research design is thus flawed. The puzzling, although significant findings from prominent authors, do not mean that the research is done. Equating, from a strict behavioural perspective, time donation and money donation is, softly spoken, a debatable approach. From one perspective, what is missing here is a unified theoretical model to support the research. We have a model explaining the inner mechanism activated when people donate money, a different model dealing with the identifiable victim within the context of helping and donating, some ideas regarding changing attitudes and behaviours throughout advertising campaigns, and a few insights about communities and tourism. 

The results presented at the end of this paper are, in fact, fully supported by this research design. This picture could not possibly be different in these circumstances.

The underlying logic of the research design must align with social cognition’s understanding of human behaviour in a social context. Adding those important perspectives referenced in the paper to this framework shall be the foundation for a solid paper. 

The data collection approach and the statistical apparatus can be improved. However, as it is now it is still a workable solution. 

As is it now, the paper is not suitable (yet) for this journal.

Reviewer 2 Report

Dear author 

The article has an attractive topic and is designed properly. However, there are some minor issues that you may need to consider. Please consider my comments below: 

1. There are some minor errors in the reference list. 

2. I recommend you to work more on the conclusion part, or it may be better to mix these two sections (discussion and conclusion). 

3. The introduction part can be developed in detail. When there is an interesting topic like this, you can have more explanations about the contribution, research gap, and importance of the topic. 

4. There is no figure. A figure may help the readers to understand and read the article faster and more effectively. 

Round 2

Reviewer 1 Report

Some of the points mentioned in the previous peer-review report were thoroughly addressed. However, what it is still not addressed is the need to picture out, for the readers, from a strict behavioural perspective, the difference between time donation and money donation. Not to mention here also the importance of the particular context in which those behaviours were performed. 

On the other hand, we have to be aware of the fact that, somehow, this idea is the cornerstone of the whole research design. If the design research is not changed, it is hard to believe that the interpretation of the data might significantly differ from the first version of the paper.

Some of the referenced papers, mainly their strong points, are not substantiated yet in the text. 

One might consider to recalibrate the paper as an exploratory study, but as an original research paper, with this design, I’m afraid it is almost impossible to be done. 
